# Improved Target Signal Source Tracking and Extraction Method Based on Outdoor Visible Light Communication Using a Cam-Shift Algorithm and Kalman Filter

**DOI:** 10.3390/s18124173

**Published:** 2018-11-28

**Authors:** Mouxiao Huang, Weipeng Guan, Zhibo Fan, Zenghong Chen, Jingyi Li, Bangdong Chen

**Affiliations:** 1School of Automation Science and Engineering, South China University of Technology, Guangzhou 510640, China; auneohuang@mail.scut.edu.cn (M.H.); auzanefan@mail.scut.edu.cn (Z.F.); auzhchan@mail.scut.edu.cn (Z.C.); aulijingyi@mail.scut.edu.cn (J.L.); 2School of Electronic and Information Engineering, South China University of Technology, Guangzhou 510640, China; eecbd_scut@mail.scut.edu.cn

**Keywords:** outdoor visible light communication (OVLC), complementary metal-oxide-semiconductor (CMOS) sensor, improved Cam-Shift algorithm, Kalman filter, robustness, real-time tracking, premise of communication

## Abstract

An improved Cam-Shift algorithm with a Kalman filter applied to image-sensor based on outdoor visible light communication (OVLC) is presented in this paper. The proposed optimized tracking algorithm is used to track and extract the region of the target signal source Light Emitting Diode (LED) that carries modulated information for data transmission. Extracting the target signal source LED area is the premise of an image-sensor-based VLC system, especially in outdoor dynamic scenes. However, most of the existing VLC studies focus on data transmission rate, visible light positioning, etc. While the actual first step of realizing communication is usually ignored in the field of VLC, especially when the transmitter (signal source LED) or the receiver (image sensor) is moving in a more complex outdoor environment. Therefore, an improved tracking algorithm is proposed in this paper, aiming at solving the problem of extracting the region of the target signal source LED accurately in dynamic scenes with different interferences so as to promote the feasibility of VLC applications in outdoor scenes. The proposed algorithm considers color characteristics and special distribution characteristics of the moving target at the same time. The image is converted to a color probability distribution map based on the color histogram of the target and adaptively adjusts the location and size of the search window based on the results obtained from the previous frame. Meanwhile, it predicts the motion state of the target in the next frame according to the position and velocity information of the current frame to enhance accuracy and robustness of tracking. Experimental results show that the tracking error of the proposed algorithm is 0.85 cm and the computational time of processing one frame is 0.042 s. Besides, results also show that the improved algorithm can track and extract the target signal source LED area completely and accurately in an environment of many interference factors. This study confirms that the proposed algorithm can be applied to an OVLC system with many interferences to realize the actual first step of communication in an image-sensor-based VLC system, laying foundations for subsequent data transmission and other steps.

## 1. Introduction

Visible light provides not only illumination but also communication, which is called visible light communication (VLC). Many applications based on VLC have been developing such as intelligent public illumination [1], indoor communication [2,3], indoor positioning [4,5], intelligent transport systems [6,7], etc. Some studies also combine VLC with other communication techniques. In Reference [8], characteristics of WiFi and VLC are described, and a practical framework is demonstrated for both technologies to coexist, further enhancing the communication performances. In the VLC field, most of the studies are limited to indoor and the outdoor applications are less explored [9]. A VLC system mainly includes two parts: a transmitter and a receiver. The transmitter modulates the Light Emitting Diode (LED)-produced light and the receiver, based on a photodiode (PD) or an image sensor, extracts the data signal from modulated light beam. A PD-based VLC system realizes communicating by receiving the light intensity of different LEDs, it costs less, and has a fast response time, which makes it possible to measure a small period of time of signal arrival; however, the main flaw of PD is that it is not stable enough, especially in an outdoor environment. In our previous works [10,11,12,13,14], we have completed a series of research projects regarding VLC-based on PD, and it can be found that PDs are sensitive to the direction of the light beam and the interference of the background light or the reflection of the surrounding objects, the communication based on PDs would be affected and even fail. Therefore, for the weakness of PDs mentioned above, an image sensor performs better in the field of OVLC applications. 

To our knowledge, studies on image-sensor-based OVLC, where LEDs are used as a transmitter and a high-speed camera is used as the receiver, such as intelligent transport systems, are mainly aiming at increasing transmission speed, reducing the BER (bit error rate), or solving other normal communication problems. In most of the existing image-sensor-based VLC studies, the transmitter (signal source) or the receiver (camera) is fixed, or the background of received images is already clean and information can be demodulated directly from these images, such as in References [15,16,17], where only signal source LEDs are left in images after adjusting the exposure time of the camera. In addition, in the field of visible light positioning (VLP) based on VLC, which is most closely related to tracking algorithm research, few studies take accuracy, robustness, and real-time ability into consideration simultaneously. In Reference [18], a MiniMax filter is used to estimate the trajectory of the terminal (receiver), which can only support low speed motion (poor real-time ability) and is merely theoretically simulated. In Reference [19], a positioning accuracy of 7.5 cm is provided while robustness (such as motion blur, background interference) is ignored. Reference [20] proposed a probability-based algorithm to track the signal source LED under a motion blur situation, whose method is similar to our previous work [4]. While, in Reference [20], only simulation experiments were carried out and there no practical experimental results are given, making it only theoretically feasible. All these works have not taken background interference into consideration, which is the main obstacle in the applications of OVLC system. However, it is worth mentioning that image-sensor-based VLC is unlike other communication techniques, which means that there will be different interferences (luminous or reflective objects) that are left in the images though adjusting exposure time, especially in an outdoor dynamic scenario. In other words, the premise of realizing communication in an image-sensor-based VLC system with different background interferences is to track the target signal source LED and extract the target area accurately and completely from received images, while it is usually ignored in the exiting studies to our knowledge. Moreover, given the various attractive advantages of complementary metal-oxide-semiconductor (CMOS) sensors [21], a CMOS sensor is used as the receiver in this paper. 

A complete and practical image-sensor-based OVLC system contains many components, including modulation of the signal source information, tracking and extraction of signal sources, features extraction and demodulation of signal source image area, image processing and denoising, etc. Each of these parts has great research value and has much room for improvement. As mentioned above, most existing reports have conducted different degrees of research on various aspects of image-sensor-based OVLC system while focus less on specialized tracking and extraction method of signal sources, which is vitally important and is the premise of communication especially in dynamic scenarios. Moreover, the core of a tracking and extraction method is the tracking algorithm, which is a fairly broad field and has a great many aspects to study and improve. Thus, an accurate, fast, stable, and practical target tracking algorithm is needed in the area of OVLC. 

Motivated by these often-overlooked but important and pressing problems, in this paper, we propose an improved tracking algorithm based on the Cam-Shift algorithm and Kalman filter to track and extract the target signal source LED from received images with interference in an outdoor dynamic scenario, aiming at promoting the feasibility of VLC applications in outdoor scenes, and filling in the gaps in this research area, the Cam-Shift algorithm converts received original images to color probability distribution maps based on the color histogram of the signal source LED by considering the target’s color characteristic and special distribution characteristic. Then, it adaptively adjusts the location and size of the search window based on the results obtained from the previous frame. At the same time, the motion state of the target in the next frame is predicted with a Kalman filter according to the position and velocity information of the current frame to enhance the accuracy and robustness of tracking. Furthermore, it locates the final position of the target signal source LED and extracts the target area. Different from our previous work [4] where a Kalman filter was used as an auxiliary correction tool to combine measurement position information via optical flow and predicted position information via Bayesian forecast by using a weighted least square method to obtain the final positioning information, in this paper, outcome of Kalman filter is used directly as measurement to update the filter when severe similar color interference or occlusion occur; otherwise, the result of Cam-Shift algorithm is used directly as a measurement to update the Kalman filter. The remainder of this paper is organized as follows. Section 2 introduces detailed theories of OVLC and the proposed algorithm. Experimental setup and results analysis are presented in Section 3. Finally, this paper is summarized in Section 4. 

## 2. Theory

### 2.1. Outdoor Visible Light Communication

The geometric model of an OVLC system based on a CMOS sensor is described in Figure 1. Data is modulated with a certain method in the transmitter and the modulated LEDs carry different information. After that, an optical signal is transmitted to the receiver and the CMOS sensor captures images after adjusting its exposure. Then the receiver extracts the target signal source LED area to extract features and demodulates these data with the corresponding method. Therefore, data transmission can be realized. 

In the proposed VLC system, a CMOS sensor is used as a receiver to realize data demodulation, whose working principle is shown in Figure 2a. According to our previous work [4], “the exposure and data readout are performed row by row, the data of one row read out immediately when the exposure of this row is finished. This is known as the rolling shutter mechanism. By the rolling shutter mechanism of CMOS sensor, turning on and off the LED light during a period of exposure would result in bright and dark stripes on the image captured by CMOS sensor,” which is shown in Figure 2b. Furthermore, in our previous works [16,21], we have described the detailed processing of modulation and demodulation, which will not be discussed in this paper. For the readers who are interested in or have questions about detailed modulation and demodulation based on a CMOS sensor, please refer to our previous reports [16,21]. 

Furthermore, in the practical application of an OVLC based on a CMOS sensor, there are three models of optical channels: (1) the transmitter is fixed and the receiver is moving, (2) the receiver is fixed and the transmitter is moving, and (3) both the transmitter and the receiver are moving. Almost all the practical application scenarios can be explained by these three models. For instance, communication between motor vehicles and fixed traffic lights corresponds to the first channel model; besides, communication between two vehicles can corresponds to all three of the channel models, as shown in Figure 1. In practical situations, both the transmitter and the receiver can be moving or stationary. Furthermore, the movement relationship between them is random, so it requires the tracking algorithm to have general applicability to be suitable for all three models of optical channels. Under different circumstances, the algorithm of tracking and extracting the target LED signal source area has different choices. When the receiver is fixed and the transmitter is moving, a background modeling method can be used to track and extract the region of the signal source, which performs well even though severe background interference occurs. In contrast, when the receiver is moving, background modeling or a frame difference method may not be applicable. Because the ego-motion of the receiver makes the background of the received images always in a state of change, leading to failure of the methods. Consequently, considering the computational complexity and to make algorithm suitable for all the three optical channel models, a more general tracking algorithm is required. Therefore, using color characteristics and special distribution characteristics, an improved Cam-Shift algorithm with a Kalman filter is proposed in this paper for signal source LED tracking and extraction to lay the foundations for the next feature extraction, data demodulation and communication. In the following parts, we introduce the basic theory and detailed processing process of the proposed algorithm and illustrate the superior performance of the proposed algorithm. 

### 2.2. Traditional Cam-Shift Algorithm

A Cam-Shift algorithm is a target tracking algorithm that uses a color histogram as the target mode. It includes three parts: calculating color projection (back projection), Mean-Shift iteration, and Cam-Shift tracking. 

#### 2.2.1. Back Projection

As the RGB (red, green and blue) color space is sensitive to changes of light intensity, a Cam-Shift algorithm converts RGB color space to HSV color space, first to decrease the impact of changes in light intensity. In the HSV color space, H, S, and V represents hue, saturation, and value respectively. Second, the H channel is spilt out as a single grayscale image to obtain its normalized histogram. Third, the value of each pixel in the image is replaced with the corresponding probability value according to the histogram of the H component. As shown in Figure 3, specific steps for calculating the back-projection are as follows:
(1)Select the LED signal source area as the tracking target and calculate its color histogram of the H component in HSV color space.(2)Obtain the hue value (H component) $h_{i,j}$ of pixel (i, j) in the image and find its corresponding interval u according to $bin\left(h_{i,j}\right)$.(3)Find the probability value of the corresponding interval u in the histogram.(4)Use this probability value as the value of pixel (i, j) of the back-projection.(5)Replace the value of each pixel in the image with its corresponding probability value.

The pixel value of each point in the original image is replaced by its color histogram according to the color histogram of the target area, which is the area of the target LED signal source in the VLC system. Then, the final projection of the image is obtained from the probability value such that the back-projection image is a color probability distribution map. In the Cam-Shift algorithm, the back projection is calculated based on the H component of the LED signal source area (the tracking target area), and the color histogram of the H component and the back-projection image are shown in Figure 4. As can be seen from Figure 4, the color of the LED signal source is different from that of the background emitting-tubes. Therefore, the target area can be located based on the back-projection image after obtaining the color histogram of the H component. 

#### 2.2.2. Mean-Shift Iteration

The Mean-Shift iteration is a robust non-parametric method for climbing density gradients to find the peak of probability distributions. Its major steps are as follows:
(1)Initialize the center and the size of the search window.(2)Calculate the center of mass of the search window:Calculate the zero moment $M_{00}$ and first moments $M_{10}$ and $M_{01}$:(1)$$\{\begin{array}{l} M_{00}=\underset{x}{\sum }\underset{y}{\sum }I\left(x,y\right)\\ M_{10}=\underset{x}{\sum }\underset{y}{\sum }xI\left(x,y\right)\\ M_{01}=\underset{x}{\sum }\underset{y}{\sum }yI\left(x,y\right) \end{array}$$where I(x,y) represents the probability value of point (x,y) in the back-projection image. Then, the center of mass $\left(x_{c},y_{c}\right)$ can be obtained:(2)$$x_{c}=\frac{M_{10}}{M_{00}},y_{c}=\frac{M_{01}}{M_{00}}$$Move the center of search window to the center of mass.(3)Exit the program when the moving distance of search window is less than a given threshold or reaches the maximum number of iterations; otherwise, return to step 2.

#### 2.2.3. Cam-Shift Tracking

The key point of a Cam-Shift algorithm is that it can adjust the size of search window and keep tracking when the size of the target area changes. The second moments $M_{11}$, $M_{20}$, and $M_{02}$ of the search window reflect the target’s orientation information. Specifically, the Cam-Shift algorithm adjusts the search window with the orientation information, which is obtained using a Mean-Shift iteration and determines the size of the search window in the next frame. $M_{11}$, $M_{20}$, and $M_{02}$ are calculated using:(3)$$\{\begin{array}{c} M_{11}=\underset{x}{\sum }\underset{y}{\sum }I\left(x,y\right)xy\\ M_{20}=\underset{x}{\sum }\underset{y}{\sum }I\left(x,y\right)x^{2}\\ M_{02}=\underset{x}{\sum }\underset{y}{\sum }I\left(x,y\right)y^{2} \end{array}$$

According to the center of mass and second moments, there are three variables a, b, and c that can be expressed as:(4)$$\{\begin{array}{c} a=\frac{M_{20}}{M_{00}}-x_{c}^{2}\\ b=\frac{M_{11}}{M_{00}}-x_{c}y_{c}\\ c=\frac{M_{02}}{M_{00}}-y_{c}^{2} \end{array}$$

Then the direction angle θ, the major axis l, and minor axis ω are obtained using:(5)$$\{\begin{array}{c} \theta =\frac{1}{2}tan^{-1}\left(\frac{2b}{a-c}\right)\\ l=\sqrt{\frac{\left(a+c\right)+\sqrt{{\left(2b\right)}^{2}+{\left(a-c\right)}^{2}}}{2}}\\ \omega =\sqrt{\frac{\left(a+c\right)-\sqrt{{\left(2b\right)}^{2}+{\left(a-c\right)}^{2}}}{2}} \end{array}$$

Thus, the initial size of the search window of the next frame is:(6)$$\{\begin{array}{c} width=max\left(l\cos\theta ,\omega \sin\theta \right)\\ height=max\left(l\sin\theta ,\omega \cos\theta \right) \end{array}$$

### 2.3. Kalman Filter

Kalman filter algorithm is an optimal linear recursive filtering algorithm based on the minimum mean square error (MMSE) and it can predict the next state according to the current state of the target. The proposed Kalman filter has three basic assumptions: (1) the modeled system is linear, (2) the noise affecting measurement is white noise, and (3) the noise follows a Gaussian distribution. The first hypothesis is that the state of the system is linear, as shown in the state prediction Equation (7) below. The latter two assumptions are that noise is white noise and follow Gaussian distribution, namely white Gaussian noise (WGN), which means that noise is random and not related to time, and it can be modeled accurately by the mean and covariance. Moreover, in the application of the Kalman filter, three kinds of motion are generally considered: dynamic motion, control motion, and random motion. Dynamic motion refers to the direct result of the system state in the previous measurement, such as uniform motion. Controlled motion is a mode of motion that is imposed on the system by some known external factors for some reason, such as accelerated motion. Random motion means random irregular movement, which can be modeled using a Gaussian model. Unlike simulation where noise can be artificially designed, the real noise is unknowable in an actual situation. Thus, in order to conform to the actual situation to the greatest extent, the motion model chooses a random model and the noise is assumed to be white noise and follows Gaussian distribution as WGN is the simplest optimal estimation of the actual noise. Then, based on the assumptions above, the state prediction equation and systematic observation equation are:(7)X(k)=A×X(k−1)+B×U(k−1)+W(k−1)
(8)Z(k)=H×X(k)+V(k)where X(k)/X(k−1) is the state value at the k/k−1 moment; Z(k) is the observation value at the k moment; and A, B, and H represents state transition matrix, control matrix, and measurement matrix, respectively. U(k−1) is the external control vector of moment k−1. W(k) and V(k) are the process noise and measurement noise, which are assumed to be white noise, independent of each other, and follow a Gaussian distribution. The corresponding covariances are Q and R, respectively.

In practice, a transition matrix A and measurement matrix H might change with each time step, but they are assumed to be constant in the proposed system to simplify the model. A control matrix B is not considered here for there is no external control vector U in the process. Moreover, the system and observation covariances Q and R are assumed to be constant. Furthermore, in the proposed system, the state matrix X is a four-dimensional vector, including the coordinates of the center position of the target signal source and velocity component in the direction of x and y. Thus, X, Z, W, V, A, H, P, Q, and R are designed to be 4×1, 2×1, 4×1, 2×1, 4×4, 2×4, 4×4, 4×4, and 2×2 matrices, respectively. 

A Kalman filter algorithm has two major parts: predict and update. It predicts the target’s current state based on the previous state of the target and obtains the predicted value of the target’s current state. Furthermore, the Kalman filter is updated with the measured value of the target of the current state. The process of predicting and updating is shown in Figure 5.

Predict (time update): Suppose that at the k−1 moment, the state vector is X(k−1|k−1) and the posterior covariance matrix is P(k−1|k−1). Then the predicted state vector X(k|k−1) and prior covariance P(k|k−1) can be forecasted based on the Kalman filter model. Because there is no external control vector in the proposed system, and based on the above definitions and assumptions, the Kalman filter time update equations are:(9)X(k|k−1)=A×X(k−1|k−1)+W
(10)$$P\left(k|k-1\right)=A\times P\left(k-1|k-1\right)\times A^{T}+Q$$

Update (measurement update): Combined with the predicted value and measured value, the optimal estimated state vector X(k|k) and posterior covariance matrix P(k|k) of the system at the k moment can be calculated using measurement update equations:(11)$$K_{g}\left(k\right)=\frac{P\left(k|k-1\right)\times H^{T}}{H\times P\left(k|k-1\right)\times H^{T}+R}$$
(12)$$X\left(k|k\right)=X\left(k|k-1\right)+K_{g}\left(k\right)\times (Z\left(k\right)-H\times X(k|k-1))$$
(13)$$P\left(k|k\right)=\left(I-K_{g}\left(k\right)\times H\right)\times P\left(k|k-1\right)$$where $K_{g}$ is called the Kalman gain. 

Furthermore, Kalman gain plays an important role in its performance and it actually has many forms under different conditions. Based on the assumptions in this paper, the most popular and the most widely used form is applied, which is named optimal Kalman gain. Other forms of Kalman gain are not applicable because they do not agree with the hypothesis of the proposed system. For simplicity, the term “Kalman gain” and the term “optimal Kalman gain” are not differentiated in this paper, which means that “optimal” can be omitted, and both are unified as Kalman gain $K_{g}$. In addition, considering the expression characteristics of Kalman gain $K_{g}$, error covariance matrix formula, and relationships between P, Q, and R, it can be concluded that under the premise that the observation covariance Q is fixed, the smaller the measurement error covariance R, the actual measurement is trusted more, while the predicted measurement is trusted less. In other words, the value of Q determines the weight of the actual measurement, and R determines the weight of the predicted measurement. Besides, what needs to be considered is that the tracking result of single Cam-Shift algorithm would be quite inaccurate when there are severe similar color disturbances, leading to relatively large errors of the final weighted result even when the weight of the actual measurement is small. Thus, in consideration of the properties of the Cam-Shift algorithm, in order to enhance robustness and minimize the error caused by interferences, values of Q should be designed to be relatively small and values of R should be designed relatively large. 

Therefore, based on the definitions and assumptions above, and under the experimental environment of the proposed system, the detailed steps of Kalman filter algorithm are as follows:
(1)Initialize the Kalman filter: The state vector of the target signal source LED is a four-dimensional vector, including the coordinate of the center position of the target and the velocity component in the directions of x and y, shown in Equation (14). Initialize the state transition matrix A, measurement matrix H, covariance matrix of motion noise Q, and covariance matrix of measurement noise R as:
(14)$$X=\left[\begin{array}{c} x\\ y\\ v_{x}\\ v_{y} \end{array}\right]$$
(15)$$A=\left[\begin{array}{cc} \begin{array}{cc} 1 & 0\\ 0 & 1 \end{array} & \begin{array}{cc} 1 & 0\\ 0 & 1 \end{array}\\ \begin{array}{cc} 0 & 0\\ 0 & 0 \end{array} & \begin{array}{cc} 1 & 0\\ 0 & 1 \end{array} \end{array}\right]$$
(16)$$H=\left[\begin{array}{cc} 1 & 0\\ 0 & 1 \end{array}\;\begin{array}{cc} 0 & 0\\ 0 & 0 \end{array}\right]$$
(17)$$Q=\left[\begin{array}{cccc} 10^{-5} & 0 & 0 & 0\\ 0 & 10^{-5} & 0 & 0\\ 0 & 0 & 10^{-5} & 0\\ 0 & 0 & 0 & 10^{-5} \end{array}\right]$$
(18)$$R=\left[\begin{array}{cc} 10^{-1} & 0\\ 0 & 10^{-1} \end{array}\right]$$

Suppose that the central position coordinate of the target signal source LED is $\left(x_{0},y_{0}\right)$ in the original frame, then initialize posterior covariance matrix P and state vector X of the target of the original frame as:(19)$$P=\left[\begin{array}{cc} \begin{array}{cc} 1 & 0\\ 0 & 1 \end{array} & \begin{array}{cc} 0 & 0\\ 0 & 0 \end{array}\\ \begin{array}{cc} 0 & 0\\ 0 & 0 \end{array} & \begin{array}{cc} 1 & 0\\ 0 & 1 \end{array} \end{array}\right]$$
(20)$$X=\left[\begin{array}{c} \begin{array}{c} x_{0}\\ y_{0} \end{array}\\ \begin{array}{c} 0\\ 0 \end{array} \end{array}\right]$$

(2)Predict: Calculate the predicted state vector X(k|k−1) and prior covariance matrix P(k|k−1) of the current frame according to Equations (9) and (10). (3)Update: The measured value Z(k) is given according to Equation (21), expressed with the central position coordinate of the target. The optimal estimated value can be obtained based on the predicted value and the measured value. Meanwhile, take the optimal estimated value as the target state of the current frame and make preparations for the next prediction.
(21)$$Z\left(k\right)=\left[\begin{array}{c} x_{k}\\ y_{k} \end{array}\right]$$(4)Return to step 2 and continue the procedure until the end of the tracking process. 

### 2.4. Improved Cam-Shift Algorithm

A traditional Cam-Shift algorithm achieves the tracking of a specific target based on a color histogram of the target area and can adaptively change the size of the search window to track the target accurately. The traditional Cam-Shift algorithm performs well under a simple background. However, it loses the target easily when the background gets complex or severe occlusion occurs. That is because under a more complicated background, there are many external factors that have an effect on the color characteristic of the target, such as the interference of the similar color, occlusion, variant posture, etc. If the traditional Cam-Shift algorithm is used under these circumstances, it will cause the wrong location of the target to be identified and eventually the target tracking fails. 

The Kalman filter algorithm is an optimal linear recursive filtering algorithm based on the minimum mean square error (MMSE) and it can predict the next state according to the current state of the target. In an OVLC system with different interferences, the target LED signal source tracking can be affected by various factors. Thus, the proposed algorithm combines traditional Cam-Shift and Kalman filter algorithms, using a Kalman filter to predict the location of the target in the next frame according to the current state information first, and then invoking a Cam-Shift algorithm that uses the predicted position as its initial searching position. The improved algorithm can not only shorten the searching path of the Cam-Shift algorithm but also reduces the effect from external factors, enhancing the robustness of the algorithm. 

A detailed procession of the proposed tracking algorithm is shown in Figure 6. First, select the target LED source signal area as the tracking target and initialize the search window. Then, use the Kalman filter algorithm and Cam-Shift algorithm separately. When there is no similar color interference or severe occlusion, the central position of the search window is used as the measurement to update the Kalman filter; otherwise, the predicted value calculated by the Kalman filter is used as the measurement. This paper uses μ and γ to determine whether the tracking is interfered by similar color or severe occlusion, respectively. The calculation formulas of μ and γ are as follows:(22)$$\{\begin{array}{c} \mu =Rect_{current}/Rect_{previous}\\ \gamma =Rect_{current}/Rect_{origin} \end{array}$$
where $Rect_{current}$ is the size of the search window of the current frame calculated using the Cam-Shift algorithm, $Rect_{previous}$ is the size of the search window of the previous frame, and $Rect_{origin}$ is the size of the initialized search window. If $\mu >threshold_{1}$, it is considered that the similar color interference is happening. If $\gamma <threshold_{2}$, it is considered that severe occlusion is occurring.

## 3. Experimental Setup and Result Analysis

### 3.1. Experimental Setup

The experimental system is shown in Figure 7, including two parts: the transmitter and the receiver. The transmitter contains direct current (DC) voltage sources, an LED, an STM32 (STMicroelectronics32) development board, and a drive circuit board. The receiver contains an industry camera, a ROS (Robot Operating System) robot, and a computer (Dell Inspiron 5557, Windows 10, 4G RAM, Intel (R) Core (TM) i5-6200U CPU @ 2.4GHz). DC voltage sources supply power to the LED and circuit boards. The LED emits a modulated optical signal to the receiver. The STM32 development board generates modulation information and the drive circuit board drives the whole transmitter circuit system. The industry camera is installed on the ROS robot, receiving modulated information from the transmitter and the ROS robot simulates the movement of a vehicle. The computer processes the received images from the industry camera with the proposed tracking algorithm to track the LED signal source in real time. The open source computer vision library (OpenCV, which was founded by Intel in 1999 and is now supported by Willow Garage. And version OpenCV 3.4.0 is used in this paper which was released on December 23, 2017) is used to complete processing process with C++ language. Specifications of the industry camera, LED, ROS robot, STM32 development board, and the drive circuit board are shown in Table 1. 

Video sequences were recorded with the industry camera introduced above to better reflect the results of the proposed algorithm under different circumstances in the applications of a CMOS-based VLC system. In the experiment, white, green, blue, and red LEDs were used to record three different videos, corresponding to the three channel models which have been introduced in 2.1. Because the proposed algorithm does not involve background modeling, such as background subtraction or a frame difference method, the three channel models are actually the same when applying our algorithm. The transmitter simulates the traffic light and the receiver, which is an ROS robot with a computer with the industry camera, simulates a moving vehicle. In the video recordings, different interferences were added, including occlusion, background interference, etc., to test the performance of the proposed algorithm in the round. 

### 3.2. Result and Analysis

There are three basic indexes judging the performance of a tracking algorithm: accuracy, robustness, and real-time performance. The accuracy of the proposed algorithm in a VLC system means the accuracy of tracking and extracting the target LED that is used as a signal source. The image processing of the tracking part is mainly conducted for selecting the transmitter, tracking the target transmitter, and extracting the signal source area from images. Extracting the target area accurately and completely is the premise of communication. Robustness is that the performance of the proposed algorithm when interference occurs, such as the sun, background light-emitting source, shielding effect, etc., in an OVLC system. Moreover, real-time performance determines whether the tracking algorithm can be applied into a practical VLC system or not; in other words, tracking and communicating would lose its meaning if the processing time of the tracking algorithm is too long, especially in a VLC system. Therefore, to demonstrate the superiority of the proposed algorithm and reflect its performance in the round, it is judged using the three aspects in this paper. 

#### 3.2.1. Accuracy Performance

In order to demonstrate the accuracy performance of the proposed tracking algorithm to the maximum extent, two hundred successive frames (101th to 300th) of video sequences (with a white LED signal source) were chosen to process. The total tracking error can be defined as: (23)$$Total\;tracking\;error=\sqrt[{2}]{{\left(x-x_{r}\right)}^{2}+{\left(y-y_{r}\right)}^{2}}$$
(24)$$error\;of\;x\;\left(y\right)=\left|value_{tracking\;algorithm}-value_{actual}\right|$$where (x,y) represents the proposed algorithm’s calculated coordinate of the target signal source LED, and $\left(x_{r},y_{r}\right)$ represents the actual coordinate of the target LED. From the proportional relationship between image coordinates and world coordinates, the actual distances can be obtained via a conversion. The tracking result is shown in Figure 8 where the blue dot represents the actual position and the red dot is the coordinate calculated by the proposed algorithm. Figure 8a is the total tracking result, Figure 8b,c is the tracking result in the *x*- and *y*-axis directions, respectively. As can be seen from the results, the performance of the tracking is good with small errors. To further illustrate the result and show the accuracy performance of the proposed algorithm in the round, Figure 9 shows the total tracking error, error of x, and error of y coordinate, where Figure 9a is the total tracking error, and the blue solid line and purple dotted line in Figure 9b represents the error of x and the error of y coordinate, respectively. The maximum total tracking error was 2.21 cm, and the average total tracking error was 0.85 cm. Furthermore, the average tracking error of x was 0.20 cm and the average error of y was 0.81 cm. There are many reasons for the tracking error, such as image noise, camera-shaking, brightness vibration of target signal source LED, etc., which can affect the back-projection image, causing the color probability distribution map to be disturbed. However, in practical applications of OVLC systems, such errors, which are only a few centimeters or within one centimeter, are acceptable. Therefore, it can be concluded that the proposed tracking algorithm has a good accuracy performance and can extract the target signal source LED area accurately and completely, which is the premise of realizing communication in a CMOS-based VLC system. 

The cumulative distribution function (CDF) is the integral of the probability density function, describing the probability of an event, the abscissa is variable and the ordinate is a probability value. Specifically, the ordinate of any point on the CDF curve indicates the probability of a variable that is less than or equal to the corresponding abscissa of the point. The CDF is defined as:(25)$$F_{X}\left(x_{0}\right)=P\left(X\leq x_{0}\right)$$

If variable is discrete, the CDF is expressed as:(26)$$F_{X}\left(x_{0}\right)=P\left(X\leq x_{0}\right)=\underset{x_{i}\leq x_{0}}{\sum }p_{i}$$where $p_{i}$ is the probability of event $x_{i}$.

The CDF is a frequently-used index for error analysis, such as in References [4,5,22]. When evaluating the tracking or positioning accuracy, the abscissa represents the error and ordinate represents the probability. Furthermore, when the ordinate value is fixed, the smaller the abscissa, along with the error value, the more accurate the algorithm performs. Usually, the ordinate value is 90% or 95% as a judging standard.

Therefore, the CDF is used in this paper to further elaborate the accuracy performance of the proposed algorithm. Shown in Figure 10, it can be seen from the curves that error of x, error of y, and total tracking error were 0.44 cm, 1.68 cm, and 1.68 cm respectively when the probability was 95%. In other words, 95% of total tracking error was less than 1.68 cm. Therefore, it can be concluded that the proposed algorithm had a good accuracy for tracking the target signal source LED. 

#### 3.2.2. Robustness Performance

Robustness performance reflects the stability of the proposed tracking algorithm, which is the first step, or rather, the premise of realizing communication in a CMOS-based VLC system. As for a CMOS-based VLC, shielding effect, background interference, and interference of similar targets are the main obstacles for a tracking algorithm and even affect the success of tracking and extracting the target signal source LED area, causing communication failure. The three influencing factors mentioned above correspond to the actual scenario, so they are added into our experimental environment to test the robustness of the proposed tracking algorithm. The video recorded with a green LED signal source is used to display experimental results in this part, and the detailed results are as follows. 

First, when the target signal source LED is complete and no interference occurs, the tracking target is selected and the search window is initialized. At the same time, the Kalman filter is initialized. As can be seen from Figure 11, the proposed algorithm tracks the target signal source LED accurately with small errors and the region of the target can be extracted successfully and completely. In other words, the proposed algorithm possesses a high tracking accuracy in this situation. 

Moreover, when the target signal source LED is masked, its valid color features information gradually decreases, making the search window shrink at the same time. The window would shrink to a minimum and stay in a local area if using the traditional Cam-Shift algorithm, causing it to be unable to track in time when the target appears completely again in future frames. Meanwhile the proposed algorithm will determine that occlusion occurs if the search window is reduced to a certain value, as shown in Figure 12. In this situation, it will use the predicted value of the Kalman filter as the measurement to update the Kalman filter and outputs the final tracking results. 

Furthermore, similar target interference and background interference occurred. A light green round LED, whose shape is the same as the target signal source LED and color is similar, entered the field of view of the industry camera. Furthermore, the background of the outdoor part appeared in the images due to reflecting sunlight, which has strong luminance and cannot be eliminated by adjusting the exposure of the industry camera. When the target signal source LED approaches the LED with similar color and severe background interference occurs, the traditional Cam-Shift algorithm would mistakenly consider the interfering LED or the background as the target and expand its search window, causing a tracking failure. The proposed algorithm can adjust its search window, shrink it, and use the Kalman filter predicted value as the measurement to update the Kalman filter when a similar color interference occurs until the target signal source LED leaves the similar background area. As can be seen from Figure 13, the extracted region contained the whole target, in other words, the proposed algorithm tracks accurately and has good robustness when interference occurred, extracting the target signal source LED area accurately, and laying foundations for communicating. 

#### 3.2.3. Real-Time Performance

The processing time of the tracking and complexity of the algorithm significantly affected the real-time performance of communication in a CMOS-based VLC system. Tracking and extracting the target signal source LED is the first step of communication. The real-time performance of communication would be greatly reduced if the running time of the tracking algorithm is too long. 

In our experiment, the transmitter was fixed and the receiver kept moving throughout the process. One hundred successive frames of each video were chosen randomly to obtain the average processing time t. The processing time of each frame of each video is shown in Table 2 so it can be calculated that t was 0.042 s. Compared to our previous work in the field of tracking and positioning, Reference [4], for instance, which uses optical flow detection and a Bayesian forecast algorithm for target tracking, its average running time was 0.162 s. Obviously, the proposed tracking algorithm in this paper was much faster. Furthermore, the accuracy of the improved Cam-Shift algorithm with a Kalman filter is basically the same as that of optical flow detection and Bayesian forecast algorithm in Reference [4], and in most practical applications, especially in an OVLC system, such errors are acceptable and it is worth mentioning that the processing time of the proposed algorithm is only one fourth of that in Reference [4], meaning that it is suitable for VLC real time communication, though it cannot forecast the next position of the target. 

Moreover, compared with other works, the proposed algorithm in this paper has a good real-time performance. References [23,24,25] can provide different degrees of positioning accuracy, which can reach decimeter-level, within ten centimeters, or even millimeter-level. However, these works focus on static positioning only and real-time ability is not considered. The balance of real-time performance, accuracy, and robustness is of vital importance and not a single one of them can be omitted in the application of VLC. In Reference [19], the computational time of each frame is only about 0.023 s for single-luminaire, which is faster than our processing time. While, reference [19] can only provide an accuracy of 7.5 cm, and the error is about 8.8 times that of ours (0.85 cm). Considering synthetically, the computational time of 0.042 s for the proposed algorithm in our paper is acceptable and has good real-time ability. Furthermore, Reference [20], whose focus was similar to ours, proposed a probability-based tracking algorithm and performed well in tracking error. However, it lacked a practical experiment and real-time analysis, meaning that it is not enough to justify its feasibility in practical application where real-time ability is required. Therefore, it can be concluded that the proposed algorithm possesses good real-time performance.

## 4. Conclusions

In this paper, an improved tracking method based on visible light communication using a Cam-Shift algorithm and a Kalman filter is proposed in order to make up for the lack of research on target signal source LED tracking and extraction in the exiting VLC reports. Obtaining the region of signal source is the premise of information demodulation in the receiver, while most studies ignore it or assume that there is no background interference in the received images and only valid data is left in these images. However, this is not the case. In an actual situation, there are other luminous or reflective object interference in images and the transmitter or the receiver can be moving or stationary, meaning the information of the signal source cannot be demodulated directly. Therefore, we proposed an improved algorithm that considered the color characteristics and special distribution characteristics of the target at the same time, converting the original image to a color probability distribution map based on the color histogram of the target and adaptively adjusting the location and size of the search window based on the results obtained from the previous frame. Meanwhile, the motion state of the target signal source LED in the next frame is predicted according to the position and velocity information of the current frame to enhance tracking robustness when interference occurs. 

As the experimental results show, the proposed algorithm possessed high accuracy and its tracking error was 0.85 cm. Furthermore, the average computational time of processing one frame was 0.042 s, meaning it had a good real-time performance. Most important of all, the improved algorithm had good robustness and the practical scenario in an image-sensor based OVLC system was taken into consideration, solving the problem of tracking and extracting the signal source area from images with different interferences, which is the premise of communication under dynamic scenes. Therefore, the proposed algorithm can be used in an image-sensor based VLC system, especially for communication in a motion state, and applied to both indoor and outdoor applications, promoting the feasibility of OVLC application and filling in gaps in this research area. 

## Figures and Tables

**Figure 1 sensors-18-04173-f001:**
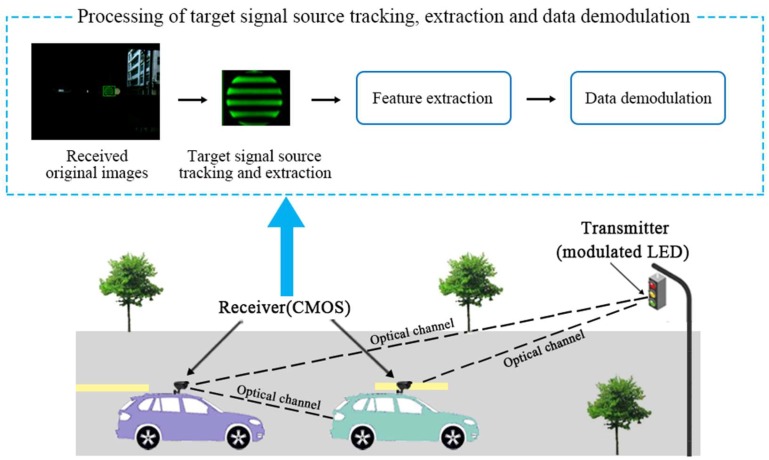
Geometric model of an OVLC system and processing in the receiver.

**Figure 2 sensors-18-04173-f002:**
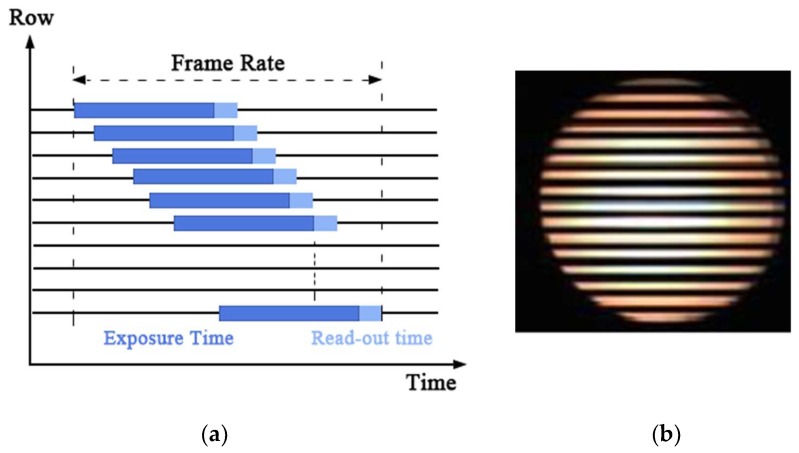
(**a**) The rolling shutter mechanism of the CMOS. (**b**) The target signal source LED area of the captured image of the CMOS sensor after a period of exposure.

**Figure 3 sensors-18-04173-f003:**
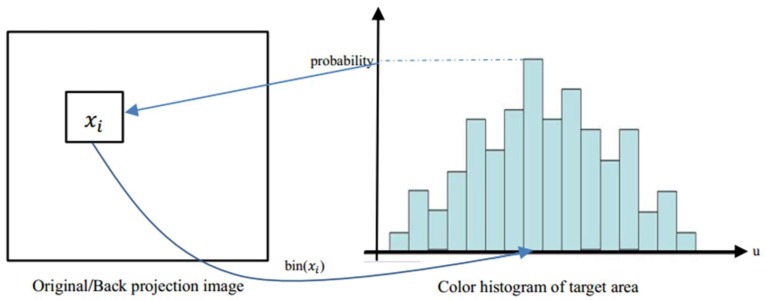
The process of obtaining the back-projection image.

**Figure 4 sensors-18-04173-f004:**
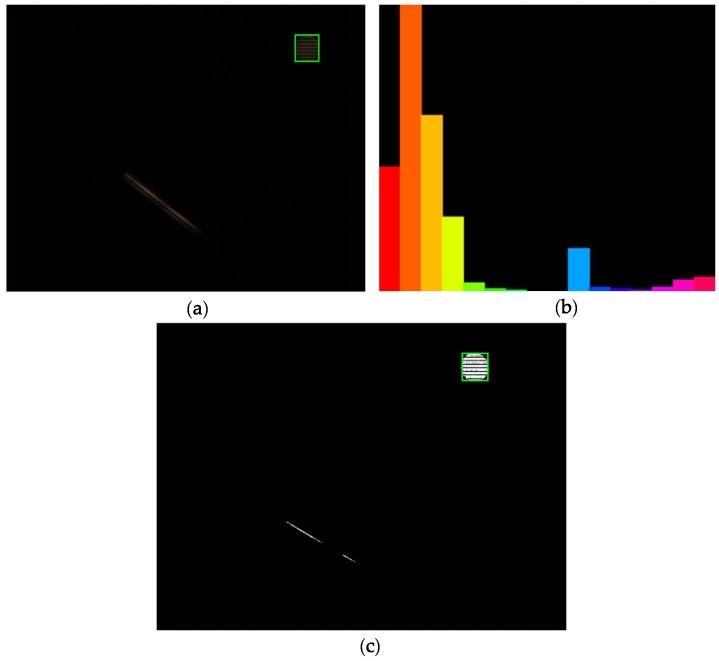
(**a**) Original image, (**b**) histogram of the H component of the target LED signal source, and (**c**) back-projection image, respectively.

**Figure 5 sensors-18-04173-f005:**
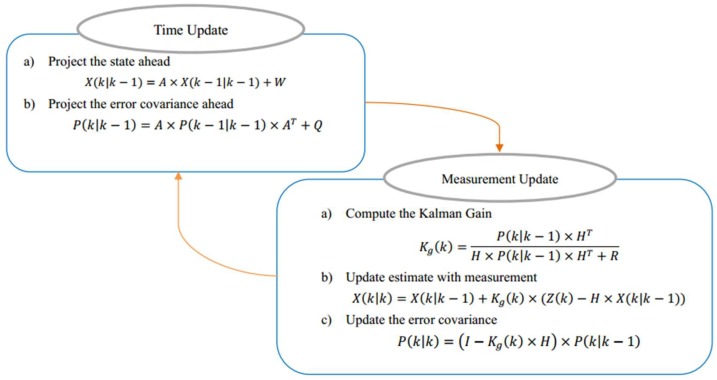
The process of predicting and updating Kalman filter.

**Figure 6 sensors-18-04173-f006:**
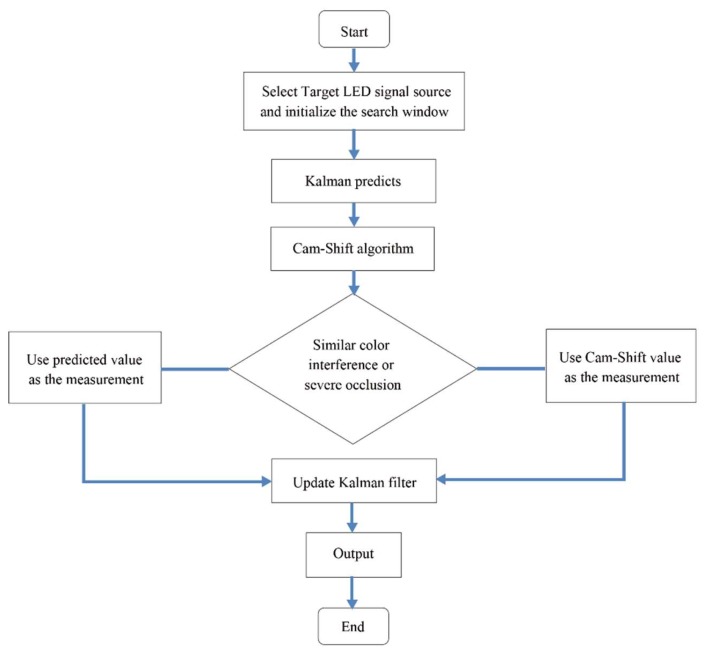
Detailed procession of the proposed tracking algorithm.

**Figure 7 sensors-18-04173-f007:**
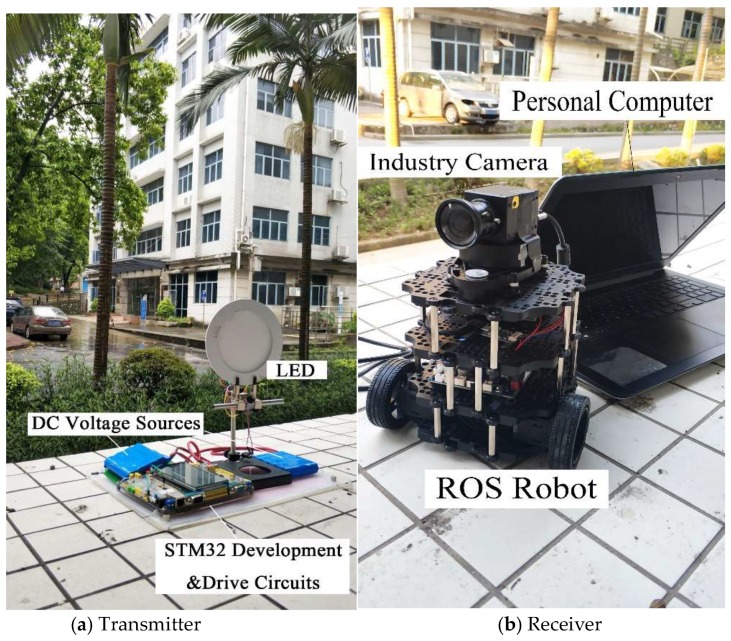
Experimental system configuration. (**a**) Transmitter, (**b**) Receiver.

**Figure 8 sensors-18-04173-f008:**
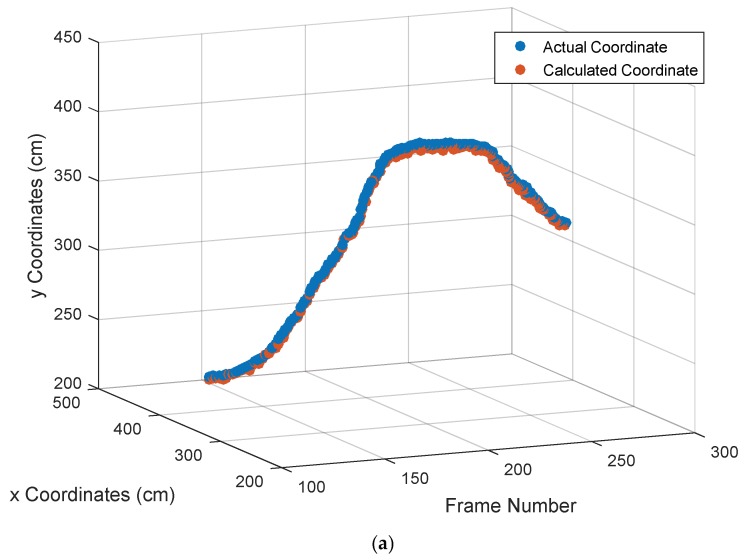
The calculated coordinate value and the actual coordinate value of the target signal source LED: (**a**) tracking result of the target signal source LED, (**b**) Tracking result of the *x* coordinate, and (**c**) tracking result of the *y* coordinate.

**Figure 9 sensors-18-04173-f009:**
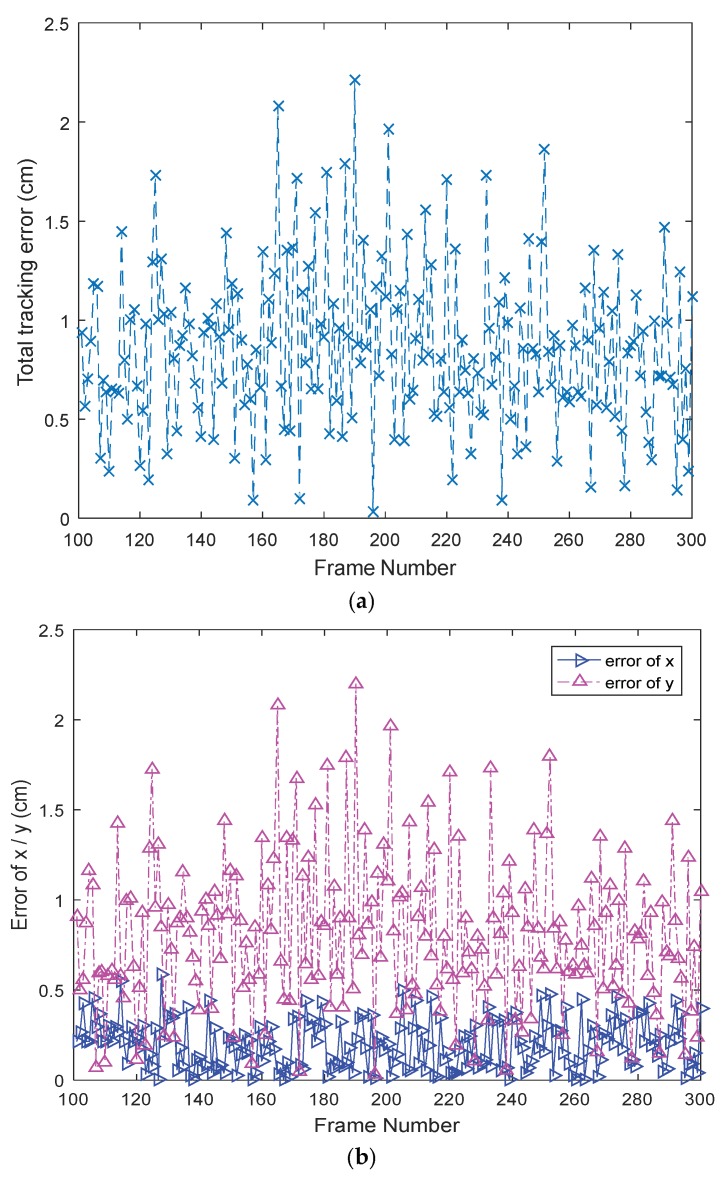
Tracking error of the target signal source LED: (**a**) total tracking error, and (**b**) tracking error of *x* and *y* coordinate.

**Figure 10 sensors-18-04173-f010:**
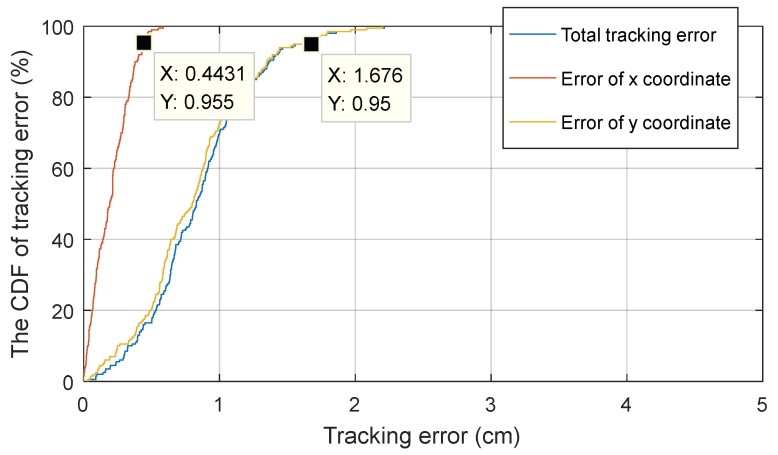
The cumulative distribution function (CDF) of the tracking error.

**Figure 11 sensors-18-04173-f011:**
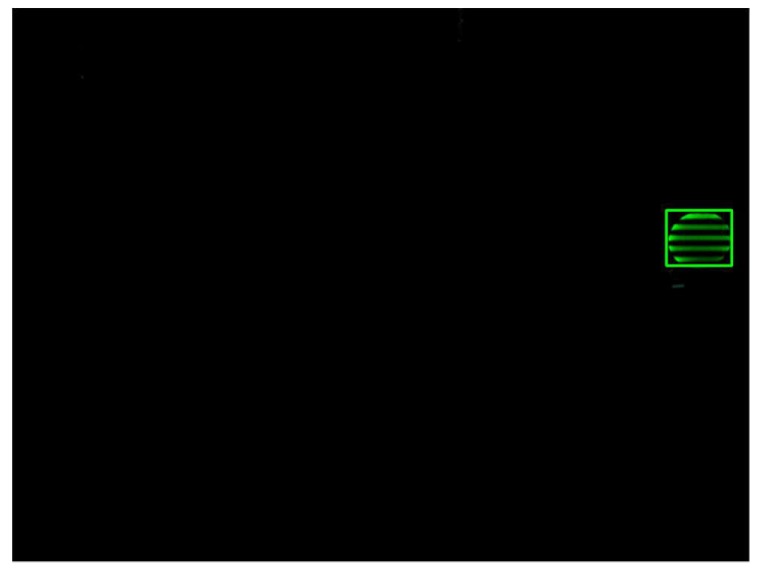
Target signal source LED selection and tracking algorithm initializing without interference.

**Figure 12 sensors-18-04173-f012:**
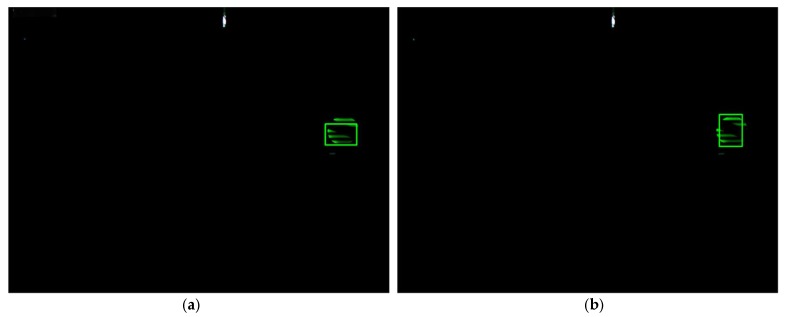
Tracking performance of the proposed algorithm when occlusion occurs: (**a**) frame number: 14, (**b**) frame number 21, (**c**) frame number 34, and (**d**) frame number 47.

**Figure 13 sensors-18-04173-f013:**
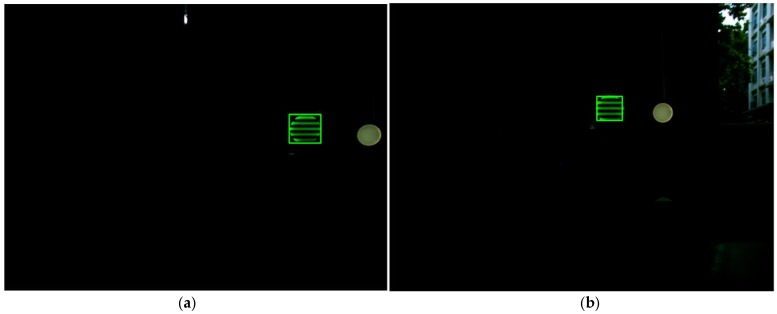
Tracking performance of the proposed algorithm under similar color interference and background interference: (**a**) frame number 63, (**b**) frame number 74, (**c**) frame number 86, and (**d**) frame number 91.

**Table 1 sensors-18-04173-t001:** Parameters of the experimental system.

Camera Specifications	
Model	MV-U300
Spectral Response Range (nm)	≈400–1030
Resolution	800 × 600
Frame Rate (FPS)	46
Dynamic Range (dB)	>61
Signal-to-noise Ratio (dB)	43
Pixel (H × V)	2048 × 1536
Pixel Size (μm²)	3.2 × 3.2
Time of Exposure (ms)	0.0556–683.8
Sensitivity	1.0 V/lux–sec 550 nm
Optical Filter	650 nm Low Pass Optical Filter
Type of Shutter	Electronic Rolling Shutter
Acquisition Mode	Successive and Soft Trigger
Working Temperatures (°C)	0–50
Support Multiple Visual Software	OpenCV, LabView
Support Multiple Systems	Vista, Win7, Win8, Win10
**LED Specifications**	
Diameter of the LED (mm)	150
Power of the LED (W)	6
The half-power angles of LED (deg($\varphi_{1/2}$))	60
**ROS Robot Specifications**	
Size (L × W × H) (mm^3^) Weight (+ SBC + Battery + Sensors) (kg) SBC (Single Board Computer)	138 × 178 × 141 1 Raspberry Pi 3 Model B
MCU	32-bit ARM Cortex® -M7 with FPU (216 MHz, 462 DMIPS)
Power connectors	3.3 V/800 mA 5 V/4 A 12 V/1 A
Power adapter (SMPS)	Input: 100–240 V, AC 50/60 Hz, 1.5 A @max Output: 12 V DC, 5 A
Actuator	Dynamixel XL460-W250
PC connection	USB
**STM32 Development Specifications**	
CPU FLASH/SRAM Power supply port	STM32F407ZGT6, LQFP144 1024 K/192 K 5 V/3.3 V
**Drive Circuit Board Specifications**	
Drive chip	DD311
Drive current (A)	0.1
Drive voltage (V)	28

**Table 2 sensors-18-04173-t002:** Results of real time performance.

LED’s Category of Videos	Processing Time of Each Frame (s)
Green signal source LED	0.042
Blue signal source LED Red signal source LED White signal source LED	0.041 0.041 0.043

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
