# Peer review of "Improved Target Signal Source Tracking and Extraction Method Based on Outdoor Visible Light Communication Using a Cam-Shift Algorithm and Kalman Filter"

_sensors, 2018, doi:10.3390/s18124173_

Round 1

Reviewer 1 Report

This article describes an approach for improving light source tracking within visible light communication (VLC) based on the integration of Kalman Filtering and the Cam-Shift algorithm. The authors address an interesting topic and provide a clear overview on the performance of VLC and the challenges yet to tackle. Although relevant factors of the system are not discussed on the paper – especially the signal modulation and the working principle of the positioning method – the provided references to their previous works shed light on them. For the evaluation of the proposed methodology, data collection was realized using a moving robot equipped with an industry camera and a LED as light source. The overall performance of the system is evaluated in terms of LED-tracking accuracy, resilience against interferences and real time capabilities.

The scientific goal of this paper is interesting and VLC is clearly a hot topic to be exploited, especially on intelligent transport systems, e.g. to enable reliable communication protocols between cars. Nonetheless, the methodology defended in this work lacks novelty and the Kalman Filter design proposed seems questionable. Additional comments and questions are as follows:

-         Writing fluency must be revised and the manuscript contains several typos which should be corrected. The use of informal expressions and contractions is not advisable. Moreover, the authors make use of the expression “What’s more” excessively.

-         The inclusion of a Kalman Filter to aid the Cam-Shift algorithm seems like a good idea. However, I strongly suggest reviewing the design of the KF, especially in regards to the system and observations variance matrices Q and R. Not only should these matrices be square, but also there should be a justification on the values of the covariance components. Moreover, beware that Eq. (10) is erroneous, as it should relate also how the system noise affect your state estimation. You can refer to [1 - 3] for a comprehensive explanation on such matter.

-         It is not addressed how the ego-motion of the vehicle affects the tracking. As mentioned during the introduction, not only could transmitter be moving but also the receiver. Unless the tracking is performed by a static observer, it is necessary to account for the self-motion. Considering that within your problem you might be tracking several light sources simultaneously, the authors might be interested in reviewing [4], where topics such as ego-motion and multi hypothesis tracking are covered.

Visible light communication and navigation is overall a fascinating topic and I would encourage the authors to continue their line of work. However, the proposed architecture needs to be revisited. Particularly, the design of the Kalman Filter is arguable and the subsections 2.3 and 2.4 are poorly explained. I would like to see a complete analysis of the approach, including an analysis of the gain obtained from the addition of the KF. However, currently I cannot recommend your paper for publication.

References:
[1] Groves, Paul D. Principles of GNSS, inertial, and multisensor integrated navigation systems. Artech house,
2013. Pages 67 – 70.

[2] Roth, Michael, Gustaf Hendeby, and Fredrik Gustafsson. "Nonlinear Kalman filters explained: A tutorial on moment computations and sigma point methods." Journal of Advances in Information Fusion 11.1 (2016): 47-70.

[3] Sola, Joan. "Quaternion kinematics for the error-state Kalman filter." arXiv preprint arXiv:1711.02508 (2017).

[4] Bar-Shalom, Yaakov, X. Rong Li, and Thiagalingam Kirubarajan. Estimation with applications to tracking and navigation: theory algorithms and software. John Wiley & Sons, 2004.

Author Response

Response to Reviewer 1 Comments

1)      Comment:

Writing fluency must be revised and the manuscript contains several typos which should be corrected. The use of informal expressions and contractions is not advisable. Moreover, the authors make use of the expression “What’s more” excessively.

Response:

Thanks for the comment. We have examined the manuscript carefully, corrected the typos and informal expressions and contractions. Moreover, the expression “What’s more” has been replaced to make the article more fluent. Sorry again for the mistakes and inappropriate expressions.

2)      Comment:

The inclusion of a Kalman Filter to aid the Cam-Shift algorithm seems like a good idea. However, I strongly suggest reviewing the design of the KF, especially in regards to the system and observations variance matrices Q and R. Not only should these matrices be square, but also there should be a justification on the values of the covariance components. Moreover, beware that Eq. (10) is erroneous, as it should relate also how the system noise affect your state estimation. You can refer to [1 - 3] for a comprehensive explanation on such matter.

Response:

Thanks very much for the comment and suggestion. As for the comment “I strongly suggest reviewing the design of the KF, especially in regards to the system and observations variance matrices Q and R”, Kalman filter algorithm is an optimal linear recursive filtering algorithm based on the minimum mean square error and it is used to estimate the state matrix                                                of a discrete controlled process that is governed by the linear stochastic difference equation

                     (1)

and observation matrix  governed by systematic observation equation

                                            (2)

Matrix  is transition matrix which relates the previous state  to the current state  and might change with each moment in practice, but it is assumed to be constant in our paper to simplify the model. Matrix  is control matrix and it is not considered here because there is no optional control vector  in our system. The measurement matrix  relates the state  to the measurement , which might change with each moment or measurement in practice, but it is regarded as constant in our paper. Moreover, the  and  represent the process noise and measurement noise, respectively. They are assumed to be independent of each other, white and with normal probability distributions, and they’re time-invariant. The corresponding system and observation covariances  and  are assumed to be constant here. Besides, posterior covariance matrix and prior covariance matrix can be defined as , .

In the proposed system, state matrix  is a four-dimensional vector, including coordinate of the centre position of the target signal source and velocity component in the direction of  and . Thus,  is a  matrix,  is a  matrix,  is a  matrix,  is a  matrix,  is a  matrix,  is a  matrix,  is a  matrix,  is a  matrix and  is a  matrix. Because there is no optional control vector in the proposed system and based on the above definitions and assumptions, Kalman filter time update equations can be obtained

                             (3)

                          (4)

and Kalman filter measurement update equations are

                                           (5)

                (6)

                              (7)

where  is called Kalman gain.

As for the comment “Not only should these matrices be square”,  and  are a  matrix and  matrix, respectively. We’re very sorry for making such mistakes, and we have corrected in the revised manuscript.

As for the comment “but also there should be a justification on the values of the covariance components”,  and  are system covariance and observation covariance, respectively, which are assumed to be constant in our experiment. Considering the expression characteristics of Kalman gain , error covariance matrix formula (4) and relationships between ,  and , it can be concluded that under the premise that the observation covariance  is fixed, the smaller the measurement error covariance , the actual measurement is trusted more, while the predicted measurement is trusted less. In other words, the value of  determines the weight of actual measurement,  determines the weight of predicted measurement.

In most experiments of Kalman filter design researches,  and  are designed as unit matrices [1]-[3]. However, tracking result of traditional Cam-Shift algorithm would be quite inaccurate when there are severe similar colour disturbances, leading to relatively large errors of the final weighted result even when the weight of actual measurement is small. Thus, in consideration of properties of Cam-Shift algorithm, values of  should be designed relatively small and values of  should be designed relatively large. After adjusting parameters and testing repeatedly, the specific values of  and  are determined. And experimental results show that the proposed system performs well with such parameters.

As for the comment “Moreover, beware that Eq. (10) is erroneous, as it should relate also how the system noise affect your state estimation”, we have examined the previous manuscript carefully and confirm that Eq. (10) (in the manuscript) is correct as the system noise was already added to Eq. (10) (in the manuscript). As can be seen in formula (4) which is the same as Eq. (10) (in the manuscript) in this response letter,  is system covariance matrix which represents the impact of system noise. And we think that what you mean is “Eq. (9) is erroneous, as it should relate also how the system noise affect your state estimation”. We examined the previous manuscript carefully and found that the system noise  is indeed missed in Eq. (9) (in the manuscript) rather than Eq. (10) (in the manuscript) under our experimental conditions and assumptions. Eq. (9) in the manuscript is the same as formula (3) in this response letter. And we have corrected it in the revised manuscript. Sorry again for such mistakes and thanks very much for your reminding.

Last but not least, it is worth emphasizing that this paper aims at promoting the feasibility of VLC application in outdoor scenes and making up the blankness in the field of target signal source tracking and extraction method based on OVLC. Because the focus of the proposed paper is OVLC instead of Kalman filter only, the specific theoretical part of Kalman filter will not be discussed too much. But indeed, the basic theoretical deduction, main implementation steps and design mentality are necessary to introduce. Thus, we have supplemented more detailed exposition especially the design and train of thought of  and  adjustment of Kalman filter in the revised manuscript. Thanks again for the comment and suggestion, and we have improved in the revised manuscript.

3)      Comment:

It is not addressed how the ego-motion of the vehicle affects the tracking. As mentioned during the introduction, not only could transmitter be moving but also the receiver. Unless the tracking is performed by a static observer, it is necessary to account for the self-motion. Considering that within your problem you might be tracking several light sources simultaneously, the authors might be interested in reviewing [4], where topics such as ego-motion and multi hypothesis tracking are covered.

Response:

Thanks for the comment and suggestion. OVLC is unlike other normal communication technologies and a complete and practical image-sensor-based OVLC system contains many components, including modulation of signal source information, tracking and extraction of signal sources, features extraction and demodulation of signal source image area, image processing and denoising, etc. Each of these parts has great research value and has much room for improvement. Most existing reports have conducted different degrees of research on various aspects of image-sensor-based OVLC system while focus less on specialized tracking and extraction method of signal sources, which is vital important and is the premise of communication especially in dynamic scenarios. In other words, all steps of data transmission and processing are based on the premise that signal sources region can be tracked and extracted accurately. Moreover, the core of tracking and extraction method is tracking algorithm which is a fairly broad field and has a great many aspects to study and improve. Thus, an accurate, fast, stable and practical target tracking algorithm is needed in the area of OVLC.

   In addition, as for the comment “As mentioned during the introduction, not only could transmitter be moving but also the receiver. Unless the tracking is performed by a static observer, it is necessary to account for the self-motion”, we have proposed three models of optical channel based on image-sensor-based OVLC in “Theory 2.1 section”, including 1) the transmitter is fixed and receiver is moving; 2) the receiver is fixed and the transmitter is moving; 3) both of the transmitter and the receiver are moving. In practical situations, both of the transmitter and the receiver can be moving or stationary. And the movement relationship between them is random, so it requires the tracking algorithm to have general applicability to be suitable for all the three models of optical channel. In the proposed paper, ego-motion effect is taken into account but it only considers its influence on basic tracking algorithm selection. For instance, background modelling method and frame difference method are not used even though they perform very well under certain conditions. Because they require that the receiver remain static, making it cannot be applied to the three models of optical channel at the same time. Moreover, this paper aims at promoting the feasibility of VLC application in outdoor scenes and making up the blankness in the field of target signal source tracking and extraction method based on OVLC, so the deeper meaning of ego-motion effect will not be discussed much in the proposed paper. Furthermore, the effect of the ego-motion of vehicles on tracking algorithm is indeed a great topic to study and this will also be one of our further research problems. Thanks very much for the suggestion and book recommendation. We have made further additions and detailed elaboration in the revised manuscript, and we will also read this book carefully and continue our research.

Reference*:

[1] Lin C L, Chang Y M, Hung C C, et al. Position Estimation and Smooth Tracking With a Fuzzy-Logic-Based Adaptive Strong Tracking Kalman Filter for Capacitive Touch Panels[J]. IEEE Transactions on Industrial Electronics, 2015, 62(8):5097-5108.

[2] Zhao Guanghui, Zhuo Song and Xu Xiaolong. Multi-target tracking method based on Kalman filter [J]. Computer Science, 2018 (8).

[3] Shi Longwei, Deng Xin, Wang Jin, et al. Multi-target tracking based on optical flow method and Kalman filter [J]. Computer applications, 2017, 37 (s1): 131-136.

Thank you for your comments concerning our manuscript and those comments are all valuable and very helpful for revising and improving our paper, as well as the important guiding significance to our researches. We have studied the comments carefully and have made corrections which we hope meet with approval. Thanks again and best regards.

Reviewer 2 Report

The manuscript presents a tracking algorithm based on Cam-Shift and Kalman filter for outdoor visible light communication. Although the manuscript is interesting and well-structured, some issues have to be addressed.

The introduction section presents some relevant references; however, the state of the art related to algorithms or methodologies for this topic is not presented (references are required). For instance, Kalman filter is also used in [4] and it is not discussed at all. Are there other algorithms? Please include and discuss about it. Why is your method better than others reported in literature? Please discuss about it. 

With a better revision of the state of the art, your contribution will be clearer. At this stage, your work seems different and new but the contribution is not such evident. Please improve it.

Some information can be referenced from [4], e.g., lines 94- 98 and Table 1.

In the Kalman filter, why is the noise assumed to be white noise and follow a Gaussian distribution in your application? How do you generate and include the white noise in the online operation?

In Equations 17 and 18, what do the numerical values represent? How were they chosen?

The Kalman gain plays an important role in its performance but nothing is said. Please, discuss about it. What is the value chosen? Why is that? What happen with others values? Provide graphic or numerical results. This is important since one of the manuscript’s novelties is the application of Kalman filter.  

For readers, the application of CDF has to be described in detail (steps, equation, etc.) in order to allow the repeatability of the tests.

In section 3.2.3, the proposal is compared with [4] (a previous work of the authors), showing some advantages. But, a qualitative/quantitative comparison with other works in literature is required.

Author Response

Response to Reviewer 2 Comments

1)      Comment:

The introduction section presents some relevant references; however, the state of the art related to algorithms or methodologies for this topic is not presented (references are required). For instance, Kalman filter is also used in [4] and it is not discussed at all. Are there other algorithms? Please include and discuss about it. Why is your method better than others reported in literature? Please discuss about it.

Response:

Thanks very much for the comment. As for the comment “the state of the art related to algorithms or methodologies for this topic is not presented”, for all we know, it is the first time that the actual premise of communication in outdoor dynamic scenarios is taken into consideration on its own and a practical and specialized signal source tracking and extraction method is proposed in the field of VLC. In other words, most of the exiting studies on image-sensor-based VLC focus on increasing transmission speed, reducing bit error rate or other normal communication problems. Besides, among most of these studies, the transmitter (signal source) or the receiver (camera) is fixed or the background of received images is already clean and information can be demodulated directly from these images, such as references [5]-[7], only signal source LEDs are left in images after adjusting the exposure time of the camera. However, in practical situation, there will be other luminous or reflective objects interference in images, causing the information of signal source cannot be demodulated directly. Thus, the premise of realizing communication in an image-sensor-based VLC system with different background interferences is to track the target signal source LED and extract it accurately and completely. This premise is vital important for behind steps in OVLC system and has great research value, and so it’s worth being singled out for discussion.

As for the comment “Kalman filter is also used in [4] and it is not discussed at all”, the usage and purpose of Kalman filter in our previous work [4] are different from those in this manuscript. In [4], Kalman filter is used to combine measurement position information via optical flow and predicted position information via Bayesian forecast. The result calculated by using Weighted Least Square Method is used as the input (measurement) of Kalman filter to update it and outputs the final positioning information (see 2.3 in reference [4]). Kalman filter is used as an auxiliary correction tool in [4]. While in the manuscript, predicted value of Kalman filter is used directly as measurement to update filter when severe similar colour interference or occlusion occur, otherwise, result of Cam-Shift algorithm is used directly as measurement to update Kalman filter. The predicted value and the tracking result of Cam-Shift are not combined to update Kalman filter, because we consider that the instability of Cam-Shift algorithm when similar colour interference occurs. The search window will expand and get rather inaccurate if similar colour objects enter the field of vision of camera. When the deviation is very large, even if the weight is small, the final fusion result would have great error.

As for the comment “Are there other algorithms? Please include and discuss about it. Why is your method better than others reported in literature”, [8] proposed a probability-based algorithm to track the signal source LED under motion blur situation, which is very similar to our previous work [4]. However, there are only simulation experiments in reference [8], making it only theoretically feasible and no actual experimental results in practical are given. In the field of visible light positioning (VLP), which is most closely related to tracking algorithm research, few studies take real-time ability, robustness and accuracy into consideration simultaneously. In [5], MiniMax filter is used to estimate trajectory of terminal (receiver), while it can only support low speed motion (poor real-time ability) and just theoretically simulated. In [6], it provides a positioning accuracy of 7.5 cm but ignores robustness (such as motion blur, background interference). All these works haven’t taken background interference into consideration, which is the main obstacle in the applications of OVLC system. Therefore, the topic and the proposed method are novel in the field of VLC. Sorry again for our lack of illustrating research innovativeness and comparison with related exiting studies, and we have corrected in the revised manuscript.

2)      Comment:

With a better revision of the state of the art, your contribution will be clearer. At this stage, your work seems different and new but the contribution is not such evident. Please improve it.

Response:

Thanks for the comment and suggestion. A complete and practical image-sensor-based outdoor visible light communication (OVLC) system contains many components, including modulation of signal information, tracking and extraction of signal sources, features extraction and demodulation of signal source image area, image processing and denoising, etc. Each of these parts has great research value and has much room for improvement. Most existing reports have conducted different degrees of research on various aspects of image-sensor-based OVLC system while focus less on specialized tracking and extraction method of signal sources, which is vital important and is the premise of communication especially in dynamic scenarios. Moreover, the core of tracking and extraction method is tracking algorithm which is a fairly broad field and has a great many aspects to study and improve. An accurate, fast, stable and practical target tracking algorithm is needed in the area of OVLC. Thus, we proposed an improved target signal source tracking and extraction method using Cam-Shift and Kalman filter to promote the feasibility of VLC application in outdoor scenes and make up the blankness in this subject. We’re very sorry for the lack of illustrating the contribution of our work, and we have further clarified the significance and innovation of the study in our revised manuscript.

3)      Comment:

Some information can be referenced from [4], e.g., lines 94- 98 and Table 1.

Response:

Thanks for the comment and suggestion. As for “lines 94-98” in the manuscript, this section introduced the basic principle of CMOS sensor, which is necessary for readers to understand this paper. Thus, this part is kept in order to facilitate readers’ basic reading and comprehension rather than referring to our previous work [4]. As for “Table 1”, not all parameters are the same as those in reference [4]. Some hardware parameters are the same but some are different and new. Hence, this part should be kept for the convenience of readers. Thanks for the suggestion and we have added reference marks to the parts in our revised manuscript.

4)      Comment:

In the Kalman filter, why is the noise assumed to be white noise and follow a Gaussian distribution in your application? How do you generate and include the white noise in the online operation?

Response:

Thanks very much for the comment. As for the comment “why is the noise assumed to be white noise and follow a Gaussian distribution”, because this is the two important assumptions of Kalman filter. Theoretically, there are three important basic assumptions in the application of Kalman filter: 1) the modelled system is linear; 2) the noise affecting measurement is white noise; 3) the noise follows Gaussian distribution. The first hypothesis is that the state of the system is linear, as shown in the state prediction equation (1) below. The latter two assumptions are that noise is white noise and follow Gaussian distribution, namely white Gaussian noise (WGN), which means that noise is random and not related to time, and it can be modelled accurately by the mean and covariance.

Moreover, in the application of Kalman filter, three kinds of motion are generally considered: dynamic motion, control motion and random motion. Dynamic motion refers to the direct result of the system state in the previous measurement, such as uniform motion. Control motion is a mode of motion that is imposed on the system by some known external factors for some reason, such as accelerated motion. Random motion means random irregular movement which can be modelled by Gauss model. Unlike simulation where noise can be artificially designed, the real noise is unknowable in actual experiment. Therefore, in order to conform to the actual situation to the greatest extent, the motion model chooses random model and the noise is assumed to be white noise and follow Gaussian distribution. Such hypothesis is the most general and agree with the actual situation best.

As for the comment “How do you generate and include the white noise in the online operation”, the proposed paper conducted a practical experiment rather than a simulation experiment, the assumed white Gaussian noise is the simulation of the actual noise. Both process noise and measurement noise are assumed to be white Gaussian noise and their covariance matrix is                                                and , respectively. Besides, assume that  and  do not vary with the state of the system, so they are initialized at the beginning and maintain as constants in the whole process. We’re very sorry that we didn’t clearly clarify the basic hypothesis and establishing conditions of Kalman filter, and we have made additions and further elaboration in the revised manuscript.

5)      Comment:

In Equations 17 and 18, what do the numerical values represent? How were they chosen?

Response:

Thanks very much for the comment. Kalman filter algorithm is an optimal linear recursive filtering algorithm based on the minimum mean square error and it is used to estimate the state matrix  of a discrete controlled process that is governed by the linear stochastic difference equation

                   (1)

and observation matrix  governed by systematic observation equation

                                              (2)

Matrix  is transition matrix which relates the previous state  to the current state  and might change with each moment in practice, but it is assumed to be constant in our paper to simplify the model. Matrix  is control matrix and it is not considered here because there is no optional control vector  in our system. The measurement matrix  relates the state  to the measurement , which might change with each moment or measurement in practice, but it is regarded as constant in our paper. Moreover, the  and  represent the process noise and measurement noise, respectively. They are assumed to be independent of each other, white and with normal probability distributions. The corresponding system and observation covariances  and  are assumed to be constant here. Besides, posterior covariance matrix and prior covariance matrix can be defined as , .

In the proposed system, state matrix  is a four-dimensional vector, including coordinate of the centre position of the target signal source and velocity component in the direction of  and . Thus,  is a  matrix,  is a  matrix,  is a  matrix,  is a  matrix,  is a  matrix,  is a  matrix,  is a  matrix,  is a  matrix and  is a  matrix. Because there is no optional control vector in the proposed system and based on the above definitions and assumptions, Kalman filter time update equations can be obtained

                             (3)

                          (4)

and Kalman filter measurement update equations are

                                              (5)

                  (6)

                                 (7)

where  is called Kalman gain.

   As for the comment “In Equations 17 and 18, what do the numerical values represent”, Equation 17 and 18 are system covariance matrix and observation covariance matrix, respectively. Besides, in the proposed system,  and  are a  matrix and  matrix, respectively. We’re very sorry that we have miswritten the order of matrices  and , which should be square. We have corrected in our revised manuscript.

   As for the comment “How were they chosen”,  and  are system covariance and observation covariance, respectively, which are assumed to be constant in our experiment. Considering the expression characteristics of Kalman gain , error covariance matrix formula (7) and relationships between ,  and , it can be concluded that under the premise that the observation covariance  is fixed, the smaller the measurement error covariance , the actual measurement is trusted more, while the predicted measurement is trusted less. In other words, the value of  determines the weight of actual measurement,  determines the weight of predicted measurement.

Moreover, in most experiments of Kalman filter design researches,  and  are designed as unit matrices [1]-[3]. However, tracking result of traditional Cam-Shift algorithm would be quite inaccurate when there are severe similar colour disturbances, leading to relatively large errors of the final weighted result even when the weight of actual measurement is small. Thus, in consideration of properties of Cam-Shift algorithm, values of  should be designed relatively small and values of  should be designed relatively large. After adjusting parameters and testing repeatedly, the specific values of  and  are determined. And experimental results show that the proposed system performs well under such parameters.

We have supplemented more detailed exposition about the design and especially the train of thought of  and  adjustment (determination of specific numerical values) of Kalman filter in the revised manuscript. Sorry again for lacking of explaining the design of Kalman filter and the selection of specific parameters, and we have improved in our revised manuscript.

6)      Comment:

The Kalman gain plays an important role in its performance but nothing is said. Please, discuss about it. What is the value chosen? Why is that? What happen with others values? Provide graphic or numerical results. This is important since one of the manuscript’s novelties is the application of Kalman filter.

Response:

Thanks very much for the comments. As for the comment “What is the value chosen? Why is that? What happen with others values? Provide graphic or numerical results”, Kalman gain has many forms under different conditions. Technically, formula (5) is just one form of Kalman gain named optimal Kalman gain, which is the most popular and the most widely used form. In the proposed paper, the system is assumed to be linear, the noise is white noise and follow Gaussian distribution. In other words, Kalman filter is used for optimization and prediction in our paper. Thus, it requires the optimization of Kalman gain . Other forms of Kalman gain  are not applicable because they do not agree with the hypothesis of our proposed system. Based on the basic assumptions of the proposed paper, the optimal Kalman gain is derived as follow.

   Define  is a priori state estimation at moment  given knowledge of the process prior to step . Define  is a posteriori state estimation at moment  given measurement . And “-” represents priori, “^” represents estimation. Then prior estimation and posteriori errors can be obtained

                                                       (8)

                                                        (9)

Then the priori estimation error covariance and posteriori error covariance are

                                                   (10)

                                                      (11)

(I) Derivation of error covariance

According to definition, derive from the error covariance ,

                                               (12)

Then put formula (6) into (12),

                                            (13)

Next put formula (2) into (13), it can be obtained

                            (14)

Because noise is not related to other terms, its covariance is zero, then

                     (15)

From properties of covariance matrix and make  = , then

                            (16)

(II) Derivation of optimal Kalman gain

To optimize  is to minimize the covariance of posteriori estimation. This is also a process to minimize the two mathematical expectation of (). Thus,

                                           (17)

 Make , then formula (16) can be written as

                          (18)

Calculate the trace of  (also formula (17)) , then derivative it with respect to  and set the result equal to zero. Therefore, optimal Kalman gain  can be obtained when  gets the minimum value.

                                                  (19)

So optimal Kalman gain  is given by

                               (20)

At this point, the proof of optimal Kalman gain  have been completed.

(III) Simplification of posteriori error covariance under optimal conditions

Multiply formula (20) by  and put the result into formula (18), it can be obtained

                                              (21)

Therefore, the proves of optimal Kalman gain  (formula (5)) and posteriori error covariance matrix (formula (7)) have been completed.

As for the comment “The Kalman gain plays an important role in its performance but nothing is said. Please, discuss about it”, Kalman gain  is shown in formula (5) introduced above, which is essentially a proportional regulator. As mentioned in Response 5), considering the characteristics of the expression of , error covariance matrix formula and relationships between ,  and , it can be concluded that the value of  determines the confidence degree of actual measurement and  determines the confidence degree of predicted measurement. We’re very sorry for the lack of describing the significance of Kalman gain clearly and the lack of emphasis on the use of optimal Kalman gain under the hypothesis in the proposed paper. We have made further additions and detailed elaboration in the revised manuscript.

Last but not least, it is worth emphasizing that this paper aims at promoting the feasibility of VLC application in outdoor scenes and making up the blankness in the field of target signal source tracking and extraction method based on OVLC. Because the focus of the proposed paper is OVLC instead of Kalman filter only, the specific theoretical part of Kalman filter will not be discussed too much, nor is Kalman gain. Kalman filter is used as a linear optimal predictor in our paper to improve tracking performance of traditional Cam-Shift algorithm, aiming at proposing an accurate, stable and fast tracking and extraction method for practical applications of OVLC. But indeed, the basic theory of Kalman filter, design mentality and significance of Kalman gain are necessary to introduced more detailed. Sorry again for the lack of elaboration of the theories above and we have improved in the revised manuscript.

7)      Comment:

For readers, the application of CDF has to be described in detail (steps, equation, etc.) in order to allow the repeatability of the tests.

Response:

Thanks very much for the comment. The cumulative distribution function (CDF) is the integral of the probability density function, describing the probability of an event and the abscissa is variable and ordinate is probability value. Specifically, the ordinate of any point on the CDF curve indicates the probability of a variable that is less than or equal to the corresponding abscissa of the point. CDF is defined as

                                            (22)

If variable is discrete, CDF is expressed

                              (23)

where  is the probability of event .

CDF is a frequently-used index for error analysis, such as [4], [9] and [10]. When evaluating the tracking or positioning accuracy, the abscissa represents error and ordinate represents probability. Besides, when the ordinate value is fixed, the smaller the abscissa, also the error value, the more accurate the algorithm performs. Usually, ordinate value is 90% or 95% as a judging standard.

The detailed definition, physical meaning, calculation steps and equation of CDF introduced above have been further complemented in our revised manuscript.

8)      Comment:

In section 3.2.3, the proposal is compared with [4] (a previous work of the authors), showing some advantages. But, a qualitative/quantitative comparison with other works in literature is required.

Response:

Thanks very much for the comment and suggestion. As for the comment “But, a qualitative/quantitative comparison with other works in literature is required”, for all we know, it is the first time that the actual premise of communication in outdoor dynamic scenarios is taken into consideration on its own and a practical and specialized signal source tracking and extraction method is proposed in the field of VLC. In other words, few exiting studies are similar to ours, and only very few of the corresponding studies can be used for direct comparison. In exiting reports in the area of VLC, visible light positioning (VLP) based on image sensor has the most requirement for target signal source tracking and extraction method. Thus, comparing with the existing VLP research to show the advantages of the proposed algorithm is meaningful and necessary.

   Section 3.2.3 is about real time ability, which is one of the most important indexes to evaluate a tracking algorithm the practical feasibility of VLC. Processing time of each frame with the proposed method in our manuscript is 0.042 s. Compared to our previous work in the field of tracking and positioning, [4] for instance, which uses optical flow detection and Bayesian forecast algorithm for target tracking and its average running time is 0.162 s. Obviously, the proposed tracking algorithm in this paper is much faster. Moreover, the accuracy of improved Cam-Shift algorithm with Kalman filter is basically the same as that of optical flow detection and Bayesian forecast algorithm in [4], in most practical applications especially in OVLC system, such errors are acceptable and it is worth mentioning that the processing time of the proposed algorithm is only one fourth of that in [4], meaning that it’s suitable for VLC real time communication though it cannot forecast the next position [4] of the target.

Furthermore, references [11]-[13] can provide different degrees of positioning accuracy, which can reach decimetre-level, within ten centimetres, or even millimetre-level. However, these works focus on static positioning only and real time ability is not considered. The balance of real time performance, accuracy and robustness is vital important and not a single one of them can be omitted in the application of VLC. In [14], the computational time of each frame is only about 0.023 s for single-luminaire which is faster than our processing time. While, [14] can only provide an accuracy of 7.5 cm, and the error is about 8.8 times that of ours (0.85 cm). Considering synthetically, the computational time of 0.042 s for the proposed algorithm in our paper is acceptable and has good real time ability. Besides, [8], whose focus is similar to ours, proposed a probability-based tracking algorithm and performs well in tracking error. However, it lacks practical experiment and real time analysis, causing that it is not enough to explain its feasibility in practical application where real time ability is required. Therefore, it can be concluded that the proposed algorithm possesses good real time performance.

Reference*:

[1] Lin C L, Chang Y M, Hung C C, et al. Position Estimation and Smooth Tracking With a Fuzzy-Logic-Based Adaptive Strong Tracking Kalman Filter for Capacitive Touch Panels[J]. IEEE Transactions on Industrial Electronics, 2015, 62(8):5097-5108.

[2] Zhao Guanghui, Zhuo Song and Xu Xiaolong. Multi-target tracking method based on Kalman filter [J]. Computer Science, 2018 (8).

[3] Shi Longwei, Deng Xin, Wang Jin, et al. Multi-target tracking based on optical flow method and Kalman filter [J]. Computer applications, 2017, 37 (s1): 131-136.

[4] Guan W, Chen X, Huang M, et al. High-Speed Robust Dynamic Positioning and Tracking Method Based on Visual Visible Light Communication Using Optical Flow Detection and Bayesian Forecast[J]. IEEE Photonics Journal, PP(99):1-1.

[5] M. Alsalami Farah, “Game theory Minimax filter design for indoor positioning and tracking system using visible light communications,” in Proc. 6th Int. Conf. Inf. Commun. Manage., Dec. 14, 2016, pp. 197–200.

[6] J. Fang et al., “High-speed indoor navigation system based on visible light and mobile phone,” IEEE Photon. J., vol. 9, no. 2, Apr. 2017, Art. no. 8200711.

[7] A. Luttman, E. Bollt, and J. Holloway, “An optical flow approach to analyzing species density dynamics and transport,” J. Comput. Math., vol. 30, pp. 249–261, 2012.

[8] Huynh P, Do T H, Yoo M. A Probability-Based Algorithm Using Image Sensors to Track the LED in a Vehicle Visible Light Communication System[J]. Sensors, 2017, 17(2):347.

[9] Cai Y, Guan W, Wu Y, et al. Indoor High Precision Three-Dimensional Positioning System Based On Visible Light Communication Using Particle Swarm Optimization[J]. IEEE Photonics Journal, 2017, PP(99):1-1.

[10] Wu Y, Liu X, Guan W, et al. High-speed 3D indoor localization system based on visible light communication using differential evolution algorithm[J]. Optics Communications, 2018, 424.

[11] R. Zhang, W.-D. Zhong, Q. Kian, and S. Zhang, “A single LED positioning system based on circle projection,” IEEE hoton. J., vol. 9, no. 4, Aug. 2017, Art. no. 7905209.

[12] J.-Y. Kim, S.-H. Yang, Y.-H. Son, and S.-K. Han, “High-resolution indoor positioning using light emitting diode visible light and camera image sensor,” IET Optoelectron., vol. 10, no. 5, pp. 184–192, Oct. 1, 2016.

[13] Md. S. Hossen, Y. Park, and K.-D. Kim, “Performance improvement of indoor positioning using light-emitting diodes and an image sensor for light-emitting diode communication,” Opt. Eng., vol. 54, no. 3, Mar. 1, 2015, Art. no. 035108.

[14] Fang J, Yang Z, Long S, et al. High Speed Indoor Navigation System based on Visible Light and Mobile Phone[J]. IEEE Photonics Journal, 2017, PP(99):1-1

Thank you for your comments concerning our manuscript and those comments are all valuable and very helpful for revising and improving our paper, as well as the important guiding significance to our researches. We have studied the comments carefully and have made corrections which we hope meet with approval. Thanks again and best regards.

Round 2

Reviewer 1 Report

Dear authors,

Thank you for your detailed replay. The paper has been properly corrected the paper and the changes input are significant and appropriate.

Some additional comments:

·        Some typos and the English fluency can still be improved, especially on the newly added paragraphs (lines 65-74: […] used to estimate the trajectory […], […] All these works have not […]. Lines 406-410: […] error analysis (REFENRECE) […]).

·        Although the Kalman Filter section has been corrected, I would suggest that it could be further improved. It might not be necessary to pose the equations of a classical linear KF (Eqs. 7 and 8) when you are already defining your KF implementation on the subsequent equations. Also, consider making use of a better mathematical expression for the size of the matrices. For instance, “[…] the vector $X \in \mathbb{R}^{4 \times 1}$ is the state estimate, $R \in \mathbb{R}^{4 \times 4}$ represents the observation noise covariance, […]”.

Author Response

1)      Comment:

Some typos and the English fluency can still be improved, especially on the newly added paragraphs (lines 65-74: […] used to estimate the trajectory […], […] All these works have not […]. Lines 406-410: […] error analysis (REFENRECE) […]).

Response:

Thanks very much for the comment and suggestion. We have examined our paper carefully and corrected the mistakes. Typos and the English fluency have been improved in the revised manuscript. Thanks again for the reminding.

2)      Comment:

Although the Kalman Filter section has been corrected, I would suggest that it could be further improved. It might not be necessary to pose the equations of a classical linear KF (Eqs. 7 and 8) when you are already defining your KF implementation on the subsequent equations. Also, consider making use of a better mathematical expression for the size of the matrices. For instance, “[…] the vector $X \in \mathbb{R}^{4 \times 1}$ is the state estimate, $R \in \mathbb{R}^{4 \times 4}$ represents the observation noise covariance, […]”.

Response:

Thanks for the comment and suggestion. As for the comment “It might not be necessary to pose the equations of a classical linear KF (Eqs. 7 and 8) when you are already defining your KF implementation on the subsequent equations”, we have carefully considered the design of the proposed KF and the structure of the whole article, and we think that the basic theory and fundamental formulas are needed to be remained. Reasons are as follows: 1) The proposed KF is based on the general linear KF. The introduction of classical linear KF helps readers understand the proposed paper more smoothly. 2) The subsequent equations are derived from the primitive equations. For the convenience of readers to understand and verify the proposed KF, giving fundamental formulas is necessary. It would be inconvenient if only references about the introduction of classical linear KF are given in the paper, especially when readers want to validate the proposed KF directly.

As for the comment “Also, consider making use of a better mathematical expression for the size of the matrices. For instance, ‘[…] the vector $X \in \mathbb{R}^{4 \times 1}$ is the state estimate, $R \in \mathbb{R}^{4 \times 4}$ represents the observation noise covariance, […]’”, the introduction is a process from general to specific, which means that when introducing matrix variables of KF, the size of the matrices is uncertain. After determining that the size of state matrix                                                is , which includes coordinates of the center position of the target signal source and velocity component in the direction of  and , sizes of the other matrices can be obtained. We think that separating the two steps of the introduction and size determination of KF matrices can help readers understand what these matrices represent and how the sizes of them are chosen, though it seems that it is a little bit verbose. Moreover, when readers want to verify the derivation of the proposed KF, centralization of definitions of the order of matrices makes it more convenient. Thanks again for the suggestions and we have further improved the KF section in the revised manuscript.

Reviewer 2 Report

All my concerns have been properly addressed. One last comment, please increase the figures' quality. 

Author Response

1)      Comment:

One last comment, please increase the figures' quality.

Response:

Thanks very much for the comment and suggestion. We examined all the 13 figures in the manuscript carefully and found that some figures’ quality indeed needed to be increased. For instance, the figure “Fig. 6 Detailed procession of the proposed tracking algorithm” is too blurry and it has been replaced with a clearer figure in the revised manuscript. Thanks again for the suggestion and we have made some adjustments to the other figures to improve their quality.
